# Exploring the Utility of Cardiovascular Magnetic Resonance Radiomic Feature Extraction for Evaluation of Cardiac Sarcoidosis

**DOI:** 10.3390/diagnostics13111865

**Published:** 2023-05-26

**Authors:** Nouf A. Mushari, Georgios Soultanidis, Lisa Duff, Maria G. Trivieri, Zahi A. Fayad, Philip M. Robson, Charalampos Tsoumpas

**Affiliations:** 1Leeds Institute of Cardiovascular and Metabolic Medicine, University of Leeds, Leeds LS2 9JT, UK; 2BioMedical Engineering and Imaging Institute, Icahn School of Medicine at Mount Sinai, New York, NY 10029, USA; 3Institute of Medical and Biological Engineering, University of Leeds, Leeds LS2 9JT, UK; 4Cardiovascular Institute, Icahn School of Medicine at Mount Sinai, New York, NY 10029, USA; 5Department of Nuclear Medicine and Molecular Imaging, University Medical Centre Groningen, University of Groningen, 9713 Groningen, The Netherlands

**Keywords:** texture analysis, feature selection, PET-MRI, imaging, machine learning

## Abstract

Background: The aim of this study is to explore the utility of cardiac magnetic resonance (CMR) imaging of radiomic features to distinguish active and inactive cardiac sarcoidosis (CS). Methods: Subjects were classified into active cardiac sarcoidosis (CS_active_) and inactive cardiac sarcoidosis (CS_inactive_) based on PET-CMR imaging. CS_active_ was classified as featuring patchy [^18^F]fluorodeoxyglucose ([^18^F]FDG) uptake on PET and presence of late gadolinium enhancement (LGE) on CMR, while CS_inactive_ was classified as featuring no [^18^F]FDG uptake in the presence of LGE on CMR. Among those screened, thirty CS_active_ and thirty-one CS_inactive_ patients met these criteria. A total of 94 radiomic features were subsequently extracted using PyRadiomics. The values of individual features were compared between CS_active_ and CS_inactive_ using the Mann–Whitney U test. Subsequently, machine learning (ML) approaches were tested. ML was applied to two sub-sets of radiomic features (signatures A and B) that were selected by logistic regression and PCA, respectively. Results: Univariate analysis of individual features showed no significant differences. Of all features, gray level co-occurrence matrix (GLCM) joint entropy had a good area under the curve (AUC) and accuracy with the smallest confidence interval, suggesting it may be a good target for further investigation. Some ML classifiers achieved reasonable discrimination between CS_active_ and CS_inactive_ patients. With signature A, support vector machine and k-neighbors showed good performance with AUC (0.77 and 0.73) and accuracy (0.67 and 0.72), respectively. With signature B, decision tree demonstrated AUC and accuracy around 0.7; Conclusion: CMR radiomic analysis in CS provides promising results to distinguish patients with active and inactive disease.

## 1. Introduction

Sarcoidosis, an inflammatory multisystem disorder of unknown origin, is characterized by the formation of non-caseating granulomas [1]. Although the disease affects the lungs in more than 90% of patients, other organs and tissues such as the heart, skin, and lymph nodes may also be affected [2]. Despite the low incidence of clinical manifestations of cardiac disease, cardiac sarcoidosis can exist as a potentially fatal disease because of associated ventricular arrhythmias. Therefore, it is crucial to identify individuals with active cardiac sarcoidosis, even if they are subclinical [3]. The diagnosis of cardiac sarcoidosis can be established with certainty by means of an endomyocardial biopsy if non-caseating granulomas are identified. However, an invasive approach poses a high degree of risk that does not yield a significant improvement in sensitivity because myocardial involvement is patchy [4].

There has been an increase in the use of non-invasive advanced imaging approaches, including cardiovascular magnetic resonance imaging (CMR) and [^18^F]fluorodeoxyglucose ([^18^F]FDG) positron emission tomography (PET). [^18^F]FDG PET imaging is effective in detecting myocardial inflammation in cardiac sarcoidosis [5]. To increase the specificity of PET imaging, it is important to follow a ketogenic diet 24 h before the scan. However, while dietary restrictions can suppress myocardial physiological glucose uptake, they may not be effective, potentially resulting in false-positive results [6,7]. On the other hand, CMR has a significant role diagnosing or screening patients with cardiac sarcoidosis. It can detect signs that might indicate disease, such as myocardial fibrosis, myocardial oedema, and perfusion defects; it is also used to assess the geometry and function of both ventricles. Late gadolinium enhancement (LGE) is a technique used to detect myocardial fibrosis in cardiac sarcoidosis which typically appears in a non-coronary distribution [3]. Presence of fibrosis, however, is not capable of determining whether the disease is active or chronic.

Using quantitative measurements can provide complementary information that may overcome limitations of non-invasive approaches [8]. Radiomic analysis is an emerging methodology that automatically extracts high dimensional features from imaging data, which can later be mined and analyzed for decision support [9]. The aim of this study is to explore the utility of radiomic analysis of LGE-CMR to separate those with active cardiac sarcoidosis, based on patchy [^18^F]FDG uptake, from those with inactive cardiac sarcoidosis, without [^18^F]FDG uptake [10]. Such an outcome may prove useful in detecting active cardiac sarcoidosis even in the presence of inconclusive or false-positive results on [^18^F]FDG-PET.

## 2. Materials and Methods

### 2.1. Ethical Approval

This study is an extension of a previous study [7] and was conducted with the approval of the Institutional Review Board at Mount Sinai (GCO # 01-1032), and all participants provided written informed consent.

### 2.2. Subject Selection

PET-CMR imaging was performed at Mount Sinai Hospital in New York, on patients with clinical suspicion of cardiac sarcoidosis based on extracardiac disease and cardiac symptoms. All subjects were treatment-naïve. As recommended by Ishida et al. [11], patient preparation required 24 h of carbohydrate abstinence and a 12 h fast before the scan. Exclusion criteria included renal dysfunction, insulin-dependent diabetes, blood glucose levels greater than 200 mg/dL, pregnancy and lactation, the presence of a cardiac pacemaker or an automatic implantable cardioverter-defibrillator, as well as failed myocardial suppression defined by high maximum target-to-background ratio (TBR_max_ > 3) [10] with widespread uptake in a non-specific pattern as determined by an expert reader in the use of PET-MR to diagnose sarcoid cardiomyopathies.

### 2.3. Imaging Protocol

Simultaneous CMR and [^18^F]FDG PET was performed on an integrated PET-CMR system (Biograph™ mMR, Siemens Healthcare, Erlangen, Germany). 5 MBq/kg of [^18^F]FDG was injected into the patients intravenously. After 10 min, thoracic PET acquisition (one-bed position centered on the heart) began and lasted for 90 min; for this study, one time window (40–100 min post injection) was reconstructed. PET images were reconstructed using the iterative ordinary Poisson ordered subset expectation maximization (OP-OSEM) with three iterations and 21 subsets on a 344 × 344 × 129 image matrix and an isotropic voxel size of 2 mm, followed by post-filtering with an isotropic 4 mm Gaussian kernel. The data obtained with PET were not respiratory-gated or ECG-gated and were not corrected for any potential motion artifacts. A 3D breath-hold Dixon-based MR image was used for attenuation correction. Simultaneously with PET imaging, CMR was performed with electrocardiograph triggering; the scan included short-axis T2 mapping and cine images covering the whole left ventricle. Approximately 15 min after injection of 0.2 mmol/kg gadolinium-based contrast agent (MultiHance, Bracco, NJ, USA), inversion-recovery fast gradient-echo LGE sequences were acquired with 8 mm slice thickness and 10 mm spacing between short-axis slices across the entire myocardium.

### 2.4. Patient Classification

Following acquisition, a single expert cardiologist evaluated the results for signs of cardiac sarcoidosis. Firstly, only subjects with LGE on CMR in a non-coronary distribution representative of cardiac sarcoidosis were selected. Subsequently, subjects with patchy [^18^F]FDG uptake on PET were classified as active cardiac sarcoidosis (CS_active_) and those who did not show any [^18^F]FDG findings were classified as inactive cardiac sarcoidosis (CS_inactive_). For this study of 148 patients scanned at the institution, thirty CS_active_ and thirty-one CS_inactive_ met these criteria.

### 2.5. Segmentation

3D slicer software (Version 4.11.2; https://www.slicer.org, accessed on 1 April 2022) was used for segmentation [12,13]. Segmentations were performed by study personnel. From the LGE-CMR images, epicardial and endocardial boundaries were drawn to define a region of interest (ROI) encompassing the entire left ventricular myocardium. Radiomic features and conventional metrics were then extracted from the ROI.

### 2.6. Feature Extraction

PyRadiomics (Version 3.0.1; Harvard Medical School, Boston, MA, USA) was used to extract six feature classes (94 features in total) from the LGE-CMR images [14]. The first-order statistical features consist of histogram (HISTO)-based properties. The features in this class are used to determine the statistical values of voxel intensities and evaluate the shape of the histogram, regardless of spatial relationships [15]. The second-order statistical features comprise features that are utilized to calculate the statistical inter-relationships between adjacent voxels and can be derived from the gray level cooccurrence matrix (GLCM) [16,17]. The higher-order statistical features can be used to extract areas with increasingly coarse texture patterns [18]. They are derived from the gray level run length matrix (GLRLM), gray level dependance matrix (GLDM), gray level size zone matrix (GLSZM), and neighboring gray tone difference matrix (NGTDM) [19]. PyRadiomics adheres to most of the image biomarker standardization initiative (IBSI) feature definitions. All parameters were maintained at their default values. Harmonization was not required for these datasets as they originated from a single scanner.

### 2.7. Statistical Analysis

Statistical analyses were undertaken using Scikit-learn software (Version 0.23.2) [20]. The Mann–Whitney U test was used to compare the individual radiomic features of the study groups. The *p*-value was adjusted using a Bonferroni correction approach for multiple tests. Given an initial significance level of 0.05, and 94 features, a *p*-value of < 0.00053 was considered to be statistically significant.

For machine learning (ML), sub-sets of radiomic features (signatures A and B) were selected using two approaches. For signature A, logistic regression classifiers were trained with individual features. Stratified five-fold cross-validation was used to determine the mean area under the curve (AUC), mean accuracy, and 95% confidence intervals (CIs). Features with AUC > 0.5 and accuracy > 0.7 were retained. Then, Spearman correlation was used to detect the correlated features and the feature with the lower AUC was removed. For signature B, principal component analysis (PCA), which reduces a large number of features into a small number of principal components (PCs), was used to identify highly correlated features and reduce feature redundancy. Components that explained 90% of the cumulative variance were retained. Both signatures were used as input to test and train the following ten ML classifiers: random forest, logistic regression, support vector machine, decision tree, Gaussian process classifier, stochastic gradient descent, perceptron classifier, passive aggressive classifier, neural network classifier and k-neighbors classifier with stratified five-fold cross-validation.

## 3. Results

### 3.1. Individual Features—Diagnostic Utility

From the univariate analysis of individual features, none of the radiomic features showed statistically significant differences on Mann–Whitney U tests between CS_active_ and CS_inactive_, with *p*-values > 0.00053. Furthermore, by measuring the effect size, the majority of the radiomic features presented small effect size values ≤ 0.5, which can be improved by increasing the sample size. The ten best-performing radiomic features based on the *p*-values were shown in Table 1. All values of radiomic features and conventional metrics are provided in Appendix A.

### 3.2. Signature Building and Machine Learning Performance

For signature A, after applying a logistic regression, only nine radiomic features had AUC > 0.5 and accuracy > 0.7. The correlated features were then removed by removing the one with the lower AUC; five were retained (Figure 1). One of the retained five features with the smallest confidence interval after correlated features were removed was GLCM joint entropy. The performance of GLCM joint entropy in distinguishing those with active disease (with [^18^F]FDG) from those with inactive disease (without [^18^F]FDG) is shown in Figure 2. Following a qualitative assessment, there were 31 CS_inactive_. Of these, 77.5% were distinguished as CS_inactive_ but with a 22.5% type I error using radiomic analysis (GLCM joint entropy). On the other side, there were 30 CS_active_ based on the PET-CMR visual assessment. The GLCM joint entropy was able to identify two-thirds of these subjects as CS_active_. For signature B, by applying PCA, six PCs were retained to explain 90% of the information and are provided in Appendix A. These signatures were used to test and train the ML classifiers.

Most of the ML classifiers showed poor performance in terms of AUC (95% CI 0.09–0.95 and 0.39–0.94) and accuracy (95% CI 0.35–0.82 and 0.36–0.82), for signature A and signature B respectively, as shown in Figure 3. For signature A, the support vector machine and k-neighbors ML classifiers had good performance with AUC (0.77 and 0.73) and accuracy (0.67 and 0.72), respectively. For signature B, the decision tree ML classifier was the only one that showed good performance with AUC and accuracy ≈ 0.7, while the random forest, Gaussian process, and passive aggressive ML classifiers had high AUCs > 0.7 but poor accuracies ≤ 0.6. The performance of the best ML classifiers of both signatures is shown in Figure 4. The values for accuracy and AUC presented in Figure 1, Figure 2, and Figure 4 are provided in Appendix A.

## 4. Discussion

This study aimed to investigate the utility of radiomic features derived from LGE-CMR images to distinguish active from inactive cardiac sarcoidosis. The univariate analysis of individual radiomic features showed that the individual radiomic features were not adequate to differentiate between patients in the CS_active_ and CS_inactive_ groups with statistical significance. However, many features in the GLCM class were among the top ten individual features, although they did not reach significance. These second-order features calculate the statistical inter-relationships between adjacent voxels [16,17]. These features inform on the spatial distribution of voxel intensities and therefore are a measure of signal heterogeneity [21,22]. It is noteworthy that these second-order features outperformed many first-order features. Given that the presence of LGE (a first-order criterion) was a pre-requisite for both CS_active_ and CS_inactive_ groups, it could be expected that first-order features would not be effective discriminators; however, the overall burden of fibrosis (and therefore LGE) would also be a likely indicator of active disease. In addition, it is interesting to note similar findings in the ML approach using signature A. About half of the features were from the GLCM class. From the uncorrelated features, GLCM joint entropy represented the best performing feature with the highest accuracy (0.72) and smallest confidence interval. The propensity for GLCM joint entropy to separate the study groups can be explained by the fact that joint entropy is a measure of the degree of randomness or variance in the pixel intensities in a given pixel’s neighborhood [23,24]. Slightly higher values related to this feature were observed in the CS_active_ group than in the CS_inactive_ group, pointing to the presence of heterogeneity, one of the characteristics of this disease [25].

A number of limitations apply to this study. It should be noted, first, that the sample size is relatively small, and further studies would be necessary to verify these findings. The small sample size may lead to overfitting and type I errors. In order to reduce the impact of this issue, Bonferroni correction and dimensionality reduction techniques were applied. Caution should be taken in interpreting our findings; of note, while the radiomic feature GLCM joint entropy showed the highest performance using logistic regression during feature selection for signature A, it was not among the top-ten features under univariate analysis, indicating that many features exhibit similar performance. Further limitations of this study include the lack of an automated segmentation, a segmentation reference, and the absence of an independent clinical gold standard based on biopsy to validate the model’s performance. Further studies are required to test the reproducibility of our findings based on radiomic features.

The new knowledge gained from this study is that radiomic analysis of LGE-CMR has the potential to identify active disease based on CMR alone. CMR-based analysis of disease activity could improve management for patients who undergo [^18^F]FDG-PET-CMR imaging and exhibit non-specific findings related to failed suppression of physiological uptake of [^18^F]FDG in the myocardium. Such non-specific findings may be present in up to 25% of PET scans [6]. However, given the limitations of the study, the results should be interpreted with caution. Moreover, the trend towards successful discrimination of CS_active_ from CS_inactive_ using second-order radiomic features such as GLCM joint entropy is encouraging and may indicate promising candidates for further evaluation. In future studies, the combined radiomic analysis of [^18^F]FDG-PET [7] and LGE-CMR data may further increase the accuracy of discriminating active from inactive disease, may be useful in determining prognosis, and may aid in clinical decision-making [26].

## 5. Conclusions

This study explored the use of radiomic analysis of LGE-CMR images in patients with cardiac sarcoidosis to distinguish between active cardiac sarcoidosis (CS_active_) and inactive cardiac sarcoidosis (CS_inactive_) based on LGE-CMR data alone. Both individual radiomic features and ML classifiers based on groups of radiomic features showed a modest ability to separate CS_active_ from CS_inactive_. GLCM joint entropy (95% CI accuracy 0.68 to 0.77; AUC 0.54 to 0.85) emerged as the individual radiomic feature with the greatest accuracy in separating CS_active_ from CS_inactive_.

## Figures and Tables

**Figure 1 diagnostics-13-01865-f001:**
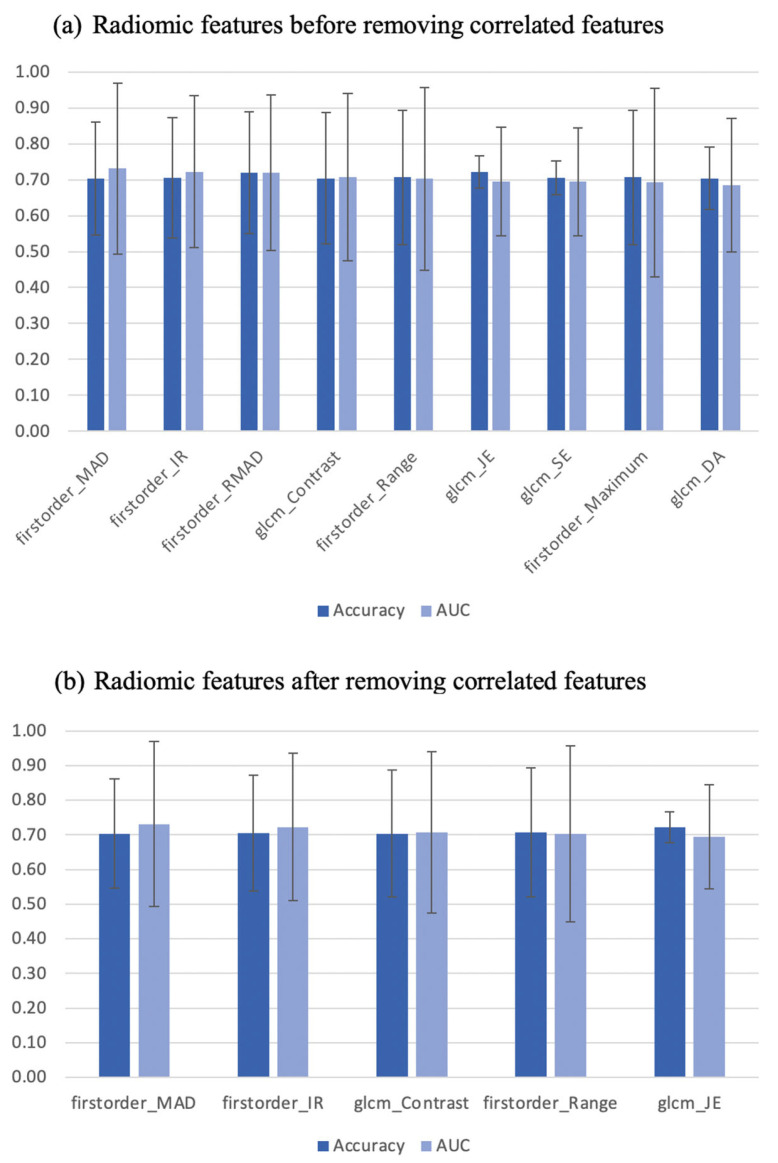
Radiomic features before and after removing correlated features with 95% confidence intervals. MAD: mean absolute deviation, IR: interquartile range, RMAD: robust mean absolute deviation, GLCM: gray level co-occurrence matrix, JE: joint entropy, SE: sum entropy, DA: difference average.

**Figure 2 diagnostics-13-01865-f002:**
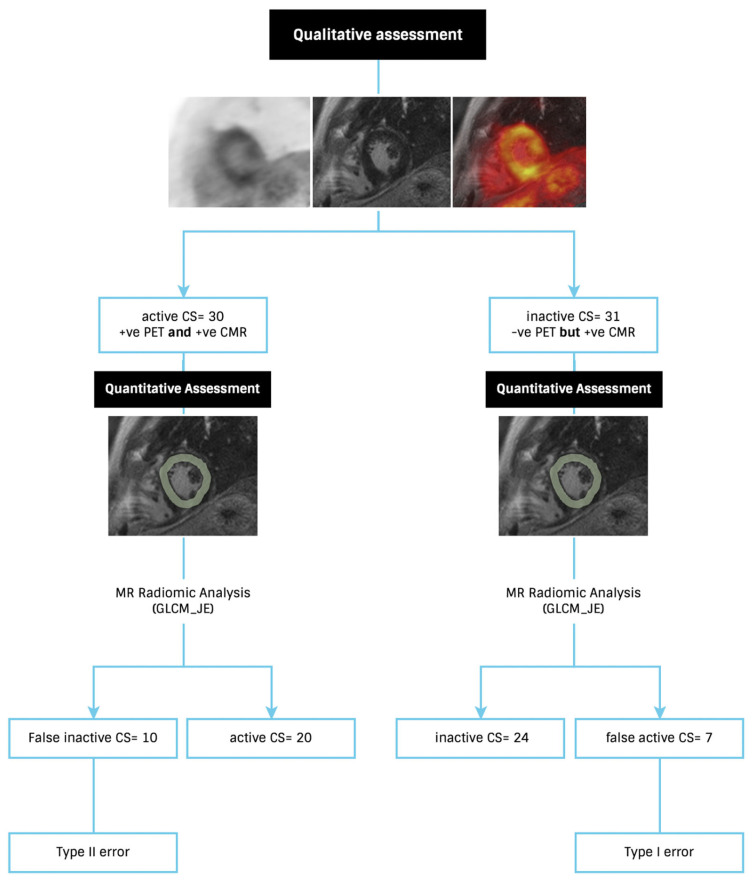
Flowchart showing the performance of the LGE-CMR based radiomic feature gray level co-occurrence matrix joint entropy (GLCM_JE) in discriminating active from inactive disease in cardiac sarcoidosis (CS).

**Figure 3 diagnostics-13-01865-f003:**
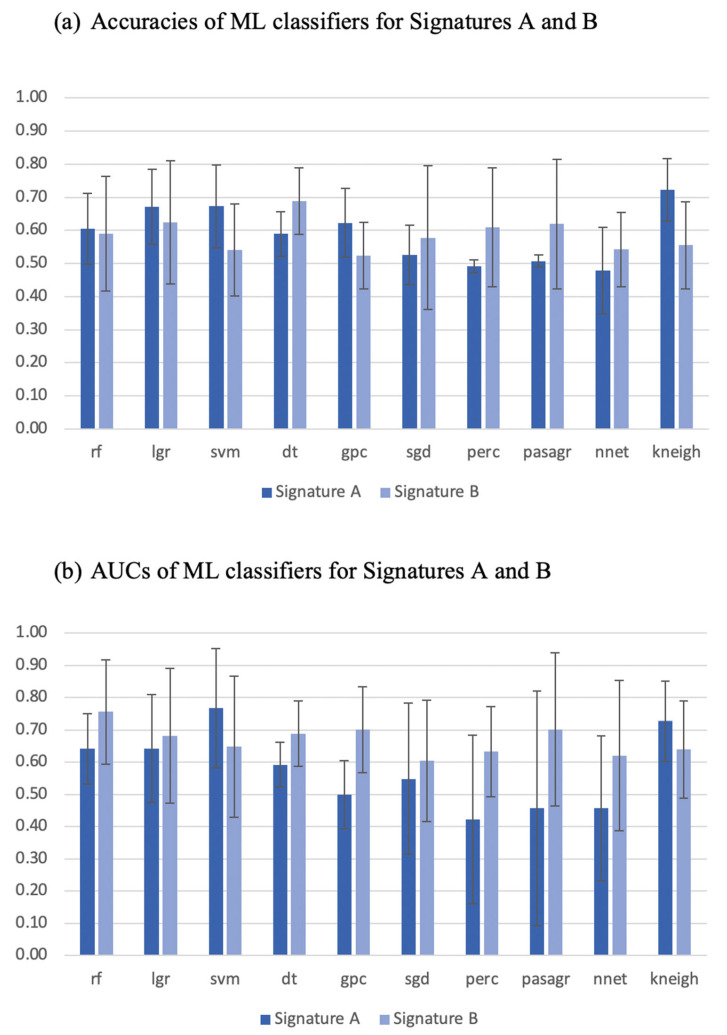
Areas under the curve (AUC) and accuracies of machine learning (ML) classifiers of both signatures with 95% confidence intervals. rf: random forest, lgr: logistic regression, svm: support vector machine, dt: decision tree, gpc: Gaussian process classifier, sgd: stochastic gradient descent, perc: perceptron classifier, pasagr: passive aggressive classifier, nnet: neural network classifier, kneigh: k-neighbors classifier.

**Figure 4 diagnostics-13-01865-f004:**
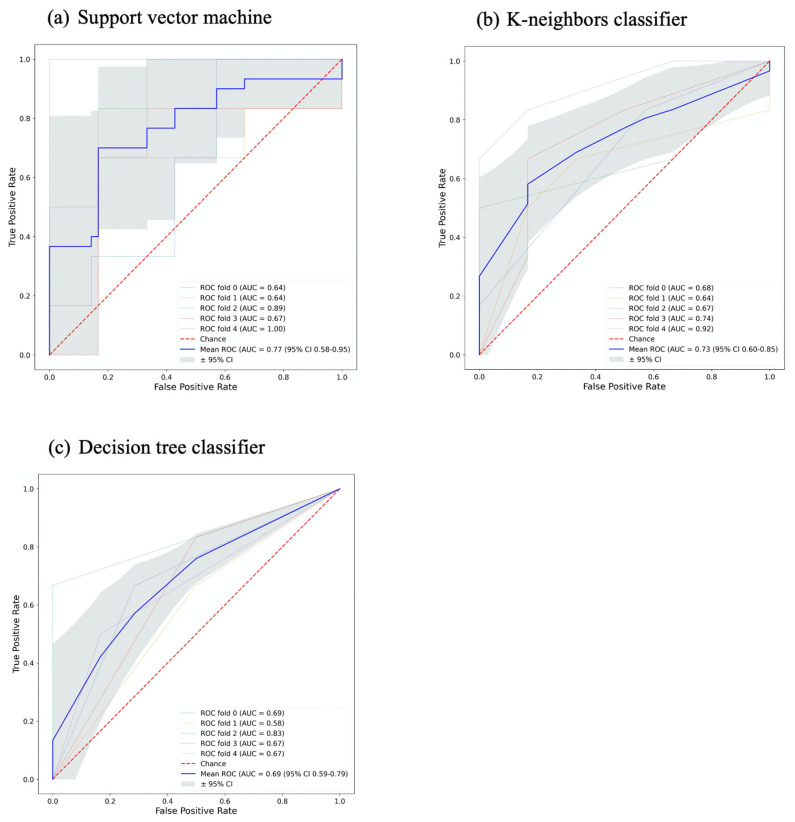
The performance of the best machine learning classifiers for both signatures, (**a**,**b**) for signature A, and (**c**) for signature B.

**Table 1 diagnostics-13-01865-t001:** The ten best-performing radiomic features based on the *p*-values. CS_active_: active cardiac sarcoidosis, CS_inactive_: inactive cardiac sarcoidosis, GLCM: gray level co-occurrence matrix, GLRLM: gray level run length matrix, GLDM: gray level dependence matrix, GLSZM: gray level size zone matrix.

Feature	CS_active_ Mean	CS_inactive_ Mean	U Statistic	*p*-Value	Effect Size
glcm_Cluster Shade	9.27	7.69	701	0.0007	0.09
glcm_Cluster Prominence	79.37	85.63	693	0.0010	0.03
firstorder_Variance	721.11	485.32	689	0.0013	0.50
glrlm_Gray Level Variance	1.36	1.01	689	0.0013	0.38
gldm_Gray Level Variance	1.21	0.83	677	0.0023	0.50
glcm_Maximal Correlation Coefficient	0.43	0.36	676	0.0024	0.60
firstorder_Mean Absolute Deviation	19.05	14.50	674	0.0026	0.75
glcm_Correlation	0.36	0.28	673	0.0028	0.61
glszm_Size Zone Non Uniformity	57.69	38.59	673	0.0028	0.81
glrlm_Run Entropy	2.98	2.81	672	0.0029	0.65

## Data Availability

The data are not publicly available due to privacy and ethical restrictions.

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
