# Peer review of "Exploring the Utility of Cardiovascular Magnetic Resonance Radiomic Feature Extraction for Evaluation of Cardiac Sarcoidosis"

_diagnostics, 2023, doi:10.3390/diagnostics13111865_

Round 1

Reviewer 1 Report

The authors test the efficacy of 94 features of first-, second-, and third-order imaging to distinguish active and inactive cardiac sarcoidosis. Methods used are generally appropriate, although not enough details is provided and the use of the opinion of a single cardiologist as a gold standard is concerning. Results are moderate, but the authors likely did not overstate their findings. 

1) Extensive language revisions must be done. Most noticeable was repeated issues with use of articles (e.g., use of "the" when not appropriate) and confusing use of plural and singular nouns (e.g., "a non-caseating granuloma forms in sarcoidosis," when in actuality sarcoidosis almost always presents with many granulomas). In some cases, this actually causes a lack of clarity. Did only one expert reader examine the images as indicated in line 78? Other small issues (not a comprehensive list): the phrase "inflammatory component" in line 46 is unclear, the purpose of dietary restrictions is unclear in line 47-48, and "scar," is probably the wrong word choice in line 54.  

2) There are some issues with references. There are many more references than is typical for this size paper; limiting to 20-25 is more appropriate. However, some critical statements do not have a reference; most notably, there should be some justification of the use of patchy FDG uptake as the distinguishing characteristic to define active and inactive cases. 

3) More details of the generation of signatures must be included. Include the number of features included at each level of signature generation in section 2.7. For example, how many were retained at step two of the generation of signature A (line 134)? When correlated features were removed, how many remained? Was the correlation coefficient a Pearson's or Spearman's, and if Pearson's, do the data satisfy the assumptions (e.g., linear relationship)? Was your logistic regression saturated/full (did it include all variable after your feature-selection steps? Include beta values and p values for the individual variables in your logistic regression. Include the loading values of the six PCs retained in signature B. 

4) Figure 1 should be redone such that panel A presents pre- and post-removal accuracy, and panel B presents pre- and post-removal AUC. This would allow for more direct comparison of the effect of removal of correlated variables. Similarly, figure 3 should be redone such that panel A presents accuracy of signatures A and B, and panel B presents AUC of signatures A and B to allow for more direct comparison of the two signatures. 

5) The final conclusion is supported by acceptable AUC and accuracy values. However, since the conclusion generally rests on the performance of one variable, GLCM joint entropy, the authors should 1) discuss why the variable did not appear as one of the ten best-performing features in their univariate (Mann-Whitney U) analyses, 2) explicitly state in the results that it is one of the five best-performing features after correlated features were removed (supplementary table 1), and 3) present the values of that variable in subjects (which they summarize, without showing the readers, in lines 209-212), especially as the supplementary table has values for the variable that are not between 3 and 4 or 1 and 2, as the text in those lines suggests. 

Author Response

Dear Reviewer, 

Thank you for all your valuable feedback. We hope the responses have addressed the comments (please see the attached file)

Reviewer 2 Report

The aim of this study is to explore the utility of radiomic analysis of late gadolinium enhancement -cardiovascular magnetic resonance (LGE-CMR) to separate those with active cardiac sarcoidosis, based on patchy [18F]fluoro-deoxyglucose (FDG) uptake, from those with inactive cardiac sarcoidosis, without FDG uptake. Such an outcome may be useful in detecting active cardiac sarcoidosis even in the presence of inconclusive or false-positive results on FDG- positron emission tomography-PET (FDG-PET).

The results of the study suggest that radiomic analysis of CMR has the potential to identify active disease based on CMR alone. CMR-based analysis of disease activity could improve patient management in this type of disease, particularly in the approximately 25% of patients undergoing PET-CMR imaging and have failed physiological myocardial uptake suppression on PET images.

Despite potential limitations (declared by the authors) the research is adequately conceived, the work is well written, the results are presented logically and in detail (textual, tabular, graphical). The topic is interesting for clinical practice.

Author Response

Dear reviewer, 

Thank you for your positive feedback. We are glad to hear that. 

Round 2

Reviewer 1 Report

The authors utilize visual features of LGE-CMR images from patients with cardiac sarcoidosis with the aim of developing a single metric or a set of metrics capable of distinguishing active and inactive sarcoidosis with high sensitivity, specificity, and accuracy. They find modest predictors. One feature, GLCM Joint Entropy, additionally demonstrated modest ability to classify patients. 

With this revision, the authors provided necessary clarity on the utilized models. In addition, grammar/language has been vastly improved.

While the study as a whole would be improved by a validation cohort to test the top-performing models, this is not necessary for this journal.